# Serum and Tissue Light-Chains as Disease Biomarkers in AL Amyloidosis

**DOI:** 10.3390/ijms26199511

**Published:** 2025-09-29

**Authors:** Alberto Aimo, Yu Fu Ferrari Chen, Michela Chianca, Francesco Mori, Angela Pucci, Vincenzo Castiglione, Veronica Musetti, Laura Caponi, Iacopo Fabiani, Giuseppe Vergaro, Maria Franzini, Michele Emdin, Daniela Cardinale

**Affiliations:** 1Health Sciences Interdisciplinary Center, Scuola Superiore Sant’Anna, 56127 Pisa, Italy; 2Fondazione Toscana Gabriele Monasterio, 56124 Pisa, Italy; 3University Hospital of Pisa, 56100 Pisa, Italy; michela.chianca@gmail.com (M.C.); francesco.mori55@gmail.com (F.M.);; 4Cardioncology Unit, Cardioncology and Second Opinion Division, European Institute of Oncology, Istituto di Ricovero e Cura a Carattere Scientifico, 20141 Milan, Italy; daniela.cardinale@ieo.it

**Keywords:** free light-chains, AL amyloidosis, plasma cell disorder, heart failure, amyloid fibrils, immunoglobulin

## Abstract

Amyloid light-chain (AL) amyloidosis is the most prevalent type of diagnosed systemic amyloidosis in Western countries, characterized by the deposition of misfolded immunoglobulin light chains (LCs) in various organs, most commonly the heart and kidneys. Circulating free LC (FLC) measurement, which can be performed by mass spectrometry or antibody-based techniques, is a crucial tool for AL amyloidosis diagnosis, risk assessment, and management. Additionally, diagnosing AL amyloidosis requires accurate detection of LC deposits in tissues. In addition to immunohistochemical techniques, mass spectrometry-based methods are now available.

## 1. Introduction

Amyloidosis is a disease caused by tissue deposition of insoluble protein fibrils composed of misfolded proteins. The term ‘amyloid’ was first introduced into human pathophysiology by Rudolf Virchow in 1854 to describe a pathologic substance initially thought to be related to starch or cellulose, but was subsequently identified as being composed of proteins [1]. Amyloid fibrils are composed of precursor proteins that undergo self-assembly into a β-sheet-rich conformation.

Proteins normally assume a well-defined three-dimensional conformation essential for their biological function and stability. Following polypeptide synthesis, proteins are vulnerable to adopting unstable, unfolded states and thus require molecular chaperones to ensure correct folding [2]. Various extracellular stressors can disrupt protein conformation, promoting the unfolding of polypeptide chains [3]. The precursor protein needs to be in an intermediate partially unfolded form in order to start amyloidogenesis [4]. While such intermediates typically refold spontaneously into their native conformation, they may occasionally misfold into aberrant, non-native structures. These misfolded proteins may be degraded via the proteasomal system or alternatively may adopt a β-sheet-rich structure, capable of self-association and subsequent polymerization into amyloid fibrils. In AL amyloidosis, this process is primarily driven by the thermodynamic instability of immunoglobulin light chains [5].

Depending on the specific amyloidogenic precursor involved, amyloid deposits in tissues give rise to distinct forms of amyloidosis. The most prevalent type, especially in Western countries [6,7] is amyloid light-chain (AL) amyloidosis, which is defined by the accumulation of free immunoglobulin light chains (FLCs) of antibodies in the tissue as a result of a plasma cell dysfunction. There are two types of clonal light-chains (LCs): lambda (found in 70–80% of individuals) and kappa (20–30%).

Structurally, light chain fibrils in AL amyloidosis are composed of full-length and often truncated light chains, which contribute to fibril heterogeneity and disease phenotype [8]. Truncated LCs, frequently missing parts of the C-terminal region, are prone to misfolding and aggregation, facilitating fibrillogenesis [9]. Amyloid deposits also commonly contain a set of amyloid-associated proteins, such as serum amyloid P component (SAP), apolipoprotein E, and apolipoprotein A-IV, which stabilize fibrils and influence tissue deposition [10]. These structural and accessory protein features are critical in determining fibril morphology, organ tropism, and the clinical course of AL amyloidosis [10].

Previously classified as a rare disorder, amyloidosis is now increasingly recognized as underdiagnosed and likely more prevalent than historically estimated. In the United States, around 4000 persons receive an AL amyloidosis diagnosis annually, with the majority of these diagnoses occurring in those between the ages of 50 and 65 [11]. Prior to 2010, the prevalence was between 8.8 and 15.5 cases per million; after 2010, it was between 40 and 58 cases per million [12].

This rise likely reflects not only increased awareness and improved diagnostic capabilities, but also growing recognition of the central role played by monoclonal light chains in the pathogenesis of the tissue damage. Quantification of FLCs is a key tool for diagnosis, management and risk assessment of AL amyloidosis. Measurement can be conducted via antibody-based assays or, alternatively, by mass spectrometry techniques. Furthermore, accurate identification of light-chain deposits in tissue is also critical for confirming the diagnosis. In addition to traditional immunohistochemical methods, mass spectrometry-based approaches are increasingly being employed for this purpose.

This review will explore the dual role of light chains in AL amyloidosis, as both drivers of disease and key biomarkers, through a structured analysis of their pathogenetic contribution, the clinical utility of measuring circulating and tissue-bound light chains, and the purposes and methodologies currently available for their detection. We will also discuss ongoing challenges and future directions in the integration of light-chain analysis into personalized diagnostic and therapeutic approaches for patients with AL amyloidosis.

## 2. Pathogenetic Role of Light-Chains in AL Amyloidosis

In AL amyloidosis, FLCs circulate in excess in the bloodstream, typically without any known physiological function. The increased production heightens the risk of fibril formation, leading to potential organ damage. However, even physiologic or near-normal levels of circulating light chains may deposit in tissues and exert pathogenic effects. FLCs can be directly cytotoxic when internalized by cells and contribute to tissue injury by forming insoluble fibrillar deposits. These deposits disrupt structural integrity and impair organ function, ultimately leading to disease manifestations driven by the failure of the affected organ [13].

Several somatic mutations have been identified in the variable domain of immunoglobulin light chains in AL amyloidosis. These mutations reduce the thermodynamic stability of the VL domain, thereby facilitating protein misfolding and aggregation into β-sheet-rich amyloid fibrils. In particular, germline gene usage appears to influence the organ tropism of LC deposition: the LV6-57 gene is associated with renal involvement, LV1-44 with cardiac amyloidosis, and KV1-33 with hepatic involvement [14]. Recent analyses have revealed additional insights into light chain mutations and germline gene usage. Sixteen germline precursor genes are significantly enriched in AL amyloidosis, with rare variants such as IGKV1-16 and IGLV1-36 showing particularly high enrichment, suggesting these may confer increased risk of amyloidogenicity [15].

A recent study [16] provides a systematic overview of the exact primary structure of the fibril proteins of AL amyloidosis patients and their respective precursor LC. In the study, they showed that mutations alter the biochemical properties of the light chains, in particular enhancing their hydrophobicity and aggregation propensity.

These findings suggest a direct link between the clonal B-cell repertoire and the pattern of organ involvement in AL amyloidosis.

Direct cell toxicity is thought to be the primary mechanism of injury exerted by amyloidogenic precursors [17]. One possible explanation for the quicker clinical development seen in AL-CA as opposed to ATTR-CA is direct injury to cardiomyocytes [18]. The mass effect of amyloid deposition, which alters tissue architecture and jeopardizes organ function, is a representation of the second damage process (Figure 1). AL amyloidosis has been referred to as a “toxic infiltrative cardiomyopathy” because of this. Cells that absorb soluble LCs are inherently poisonous [19,20]. Apoptosis, oxidative stress, mitochondrial dysfunction, poor calcium handling and contractility, aberrant autophagy, and lysosomal dysfunction are among the alterations that have been seen in animal models of light-chain cardiotoxicity [21,22]. In addition, recent studies have shown that amyloidogenic light chains activate intracellular stress pathways such as p38-MAPK and impairs autophagy through downregulation of TFEB, a master regulator of lysosomal function. These effects were attenuated by pharmacological inhibition of p38 or stimulation of autophagy, suggesting potential therapeutic targets for mitigating LC-induced cardiotoxicity [22].

The detrimental effects of amyloidogenic light chains have been investigated in the two predominant types of heart cells: cardiomyocytes [24] and fibroblasts [25]. Studies demonstrated that amyloidogenic LCs interact with proteins and other biological components, possibly by causing them to malfunction or sequester. In particular, amyloidogenic LCs seem to interfere with important metabolic pathways in mitochondria by interacting with peroxisomal ACOX1 (i.e., the first enzyme of the fatty acid beta-oxidation pathway) and OPA1 (optic atrophy 1-like protein), which is found in the inner mitochondrial membrane [26], thereby impairing key metabolic pathways. Furthermore, the pathophysiology and organ tropism of amyloidosis seem to be significantly influenced by the extracellular matrix (ECM) [27]. Exposure of cells to amyloidogenic proteins induces alterations in the secretion and processing of ECM components, with fibril deposition occurring in tight spatial association with collagen and glycosaminoglycans.

## 3. The Role of Monoclonal Light-Chains in Diagnosis, Risk Prediction and Response Assessment

FLC test, in combination with serum protein electrophoresis and serum and urine immunofixation, is essential for the accurate diagnosis of AL amyloidosis [28]. To guarantee efficient utilization of all the produced heavy chains, LCs are synthesized with a physiological excess of approximately 500 mg daily [29]. In the bloodstream, FLCs exist both as monomers and dimers. Interestingly, lambda FLCs are primarily found as dimers, while kappa FLCs are primarily found as monomers. Serum immunofixation (S-IF) can be used to further characterize monoclonal serum immunoglobulins, which may appear as an abnormal peak on serum protein electrophoresis (ELP). On the other hand, when only an excessive amount of monoclonal FLCs are produced, the ELP pattern typically remains unaltered [30]. AL amyloidosis is distinguished by comparatively modest levels of circulating monoclonal proteins in contrast to other plasma cell diseases [31]. These proteins can be identified in approximately 90–95% of patients using combined serum and urine immunofixation [32]. In contrast, serum protein electrophoresis has a considerably lower sensitivity, detecting the monoclonal component in only about half of cases [33].

The development of automated serum FLC assays has revolutionized the diagnosis and monitoring of plasma cell disorders by significantly improving the quantification of serum FLC levels, the kappa/lambda ratio, and the ratio between involved and uninvolved FLC (i/uFLC) [34].

Using the Freelite^®^ assay, the kappa/lambda ratio in people with normal kidney function falls between 0.26 and 1.65. The reticuloendothelial system becomes more crucial for FLC elimination in individuals with chronic renal disease, and the same ratio using the same assay can raise up to three times (range 0.37–3.10) (Figure 2) [35]. A normal kappa/lamba FLC ratio provides a 100% negative predictive value for AL-CA in patients with cardiac disease [36].

FLC measurement is also a crucial parameter for the follow-up of individuals already diagnosed with the disease. In the study, of amyloidosis, there has been a growing interest in the difference between the concentration of uninvolved FLCs (dFLC) and involved FLCs, which is caused by clonal growth. One of the best indicators of overall survival in AL amyloidosis is dFLC (as determined by the Freelite^®^ assay); the prognostic MAYO2012 score includes the 180 mg/L cut-off [37]. Furthermore, a >50% decrease in dFLC or a reduction to <40 mg/L are important criteria of response to treatment [38]. A later comparison study found that using N Latex FLC to assess FLCs resulted in a lower predictive cut-off for dFLC (165 mg/L) [20]. Even though the N Latex assays have similar diagnostic sensitivity and prognostic value, they cannot be utilized for disease staging or evaluating therapy response since studies that incorporate dFLC into staging systems and response criteria are based on the Freelite^®^ assay. The oligomeric form of FLCs has also been linked to differences between these two approaches [39,40]. Indeed, it has recently been shown that the N Latex reagent strongly recognizes the lambda FLC monomers, while the Freelite^®^ test better detects lambda FLC dimers connected by an inter-chain disulphide bridge [41]. More recent evidence [42] demonstrated that the calibrators of the Freelite and N-Latex FLC assays differ significantly in their content and reactivity. Specifically, they found that the Freelite calibrator contains both FLC monomers and dimers, with a dimer/monomer ratio approximately equal to 1. In contrast, the N-Latex FLC calibrator consists solely of FLC dimers. This disparity in calibrator composition contributes to the observed differences in assay performance. These findings underscore the importance of calibrator composition in determining the immunoreactivity of FLC assays and highlight the challenges in achieving standardization across different testing platforms.

Therefore, differences in the epitopes recognized by the assays may introduce additional bias, precluding their interchangeable use in clinical practice.

## 4. Methods to Measure Circulating Light-Chains

The serum FLC can now be measured using five commercial diagnostic tests: the Freelite^®^, N Latex FLC, Diazyme, Seralite^®^, and Sebia FLC assays. Each technique relies on antibodies that can recognize particular FLC epitopes. Since they appear to be only noticeable when the LCs are truly free and are invisible while attached to heavy chains, these are referred to as “hidden” [43]. Table 1 summarizes the main characteristics of these assays.

The Freelite^®^ assay (The Binding Site Group Ltd., Birmingham, UK) was the first method available on automated instruments and remained the only option for a decade. It is considered the reference assay in the International Myeloma Working Group (IMWG) recommendations [44,45]. Polyclonal antibodies against kappa and lambda light chains are obtained by immunizing sheep with urinary light chains from patients with monoclonal proteinuria (Bence–Jones proteins). These antibodies are subsequently purified through affinity techniques, in which they are exposed to intact immunoglobulins, and only those showing specific reactivity are retained. They are coated onto polystyrene latex particles and measured by nephelometry or turbidimetry. Calibration uses human serum with defined FLC content, and reference intervals are provided for the BN II System (Siemens Healthineers Diagnostics GmbH, Marburg, Germania, nephelometric technique) [45]. Limitations include variability across different instruments even when the same reagents are used, as well as underestimation due to antigen excess. Thus, while the Freelite^®^ assay enables absolute quantification, its accuracy is reduced at very high concentrations or after dilution [45,46,47]

The N Latex FLC assays (Siemens Healthineers Diagnostics GmbH, Marburg, Germany) are latex-based nephelometric assays designed specifically for the Siemens BN II systems. They employ murine monoclonal antibodies recognizing epitopes in the constant domain of FLCs, with detergents added to reduce nonspecific binding. Calibration is performed with purified polyclonal FLCs in buffered solution containing human albumin, and values are standardized against Freelite^®^. In AL amyloidosis this approach is considered valid, as quantification levels between the assays are highly concordant [20]. In contrast, discrepant results have occasionally been reported in multiple myeloma [48]. Comparative studies have shown that while the two assays demonstrate similar clinical utility, they are not analytically equivalent and therefore should not be used interchangeably for patient monitoring [49,50,51]. Accordingly, N Latex assays provide quantitative results, yet cross-platform equivalence remains an issue [52].

The Diazyme FLC assays (Diazyme Laboratories Inc., Poway, CA, USA) use latex-enhanced immunoturbidimetry with rabbit polyclonal antisera. Two comparative studies against Freelite^®^ reported acceptable agreement for kappa light chains but marked discrepancies for lambda light chains [53,54]. So, although the method yields quantitative values, reliability is reduced for lambda measurements, particularly at higher concentrations

The Seralite^®^ assay (Abingdon Health/Sebia, Evry, France) enables simultaneous determination of kappa and lambda FLCs using a competitive inhibition mechanism. Each test strip incorporates immobilized FLCs that compete with patient FLCs for binding to gold nanoparticle-conjugated monoclonal antibodies. The signal, read by the ADxLR5 device, is inversely proportional to analyte concentration [55]. This design reduces the risk of antigen excess effects. No significant differences are found in median FLC values compared to the Freelite^®^ test, although the kappa/lambda ratio of Seralite^®^ has a wider range [55]. Hence, Seralite^®^ is quantitative, with better performance in high-concentration settings, but interpretation of ratios may differ.

The Sebia enzyme-linked immunosorbent assay (ELISA) (Sebia, Evry, France) uses a sandwich format with rabbit polyclonal antibodies, with the secondary antibody labeled by horseradish peroxidase [56]. It has been fully automated on the AP22 ELITE processor. The assay correlates well with Freelite^®^ and, due to its broader dynamic range, reduces the need for repeat dilutions. However, biases emerge at higher concentrations [57]. This approach therefore provides robust quantitative measurement with improved linearity compared to other immunoassays.

Overall, serum FLC measurement significantly increases sensitivity in screening panels for AL amyloidosis, leading to its inclusion in IMWG recommendations since 2009 [58,59].

Despite improvements, immunoassays remain prone to technical limitations such as the hook effect. Alternative technologies are being explored to improve precision. Mass spectrometry (MS) has become particularly attractive with advances in accessibility and instrumentation. By targeting the unique amino acid sequence in the complementarity-determining region (CDR), MS can provide both identification and accurate quantification of monoclonal light chains.

The clonotypic peptide approach requires prior sequencing of the clonal light chain to monitor unique peptides by LC–MS/MS [60,61]. Although technically demanding, once established it offers highly sensitive and accurate quantification, particularly for measurable residual disease. By contrast [59], the miRAMM (monoclonal immunoglobulin rapid accurate mass measurement) technique does not require prior sequence knowledge [62]. Immunoglobulins were first enriched from samples, then reduced to release free light chains, which were subsequently subjected to LC-ESI-TOF MS analysis. The monoclonal light chain appeared as a single peak when the mass spectra of multiply charged light chains were deconvoluted to determine their molecular mass. The distinct *m*/*z* distribution of their constant sections allowed for the differentiation of kappa and lambda light chains. Without requiring chromatography, the same team that created this initial technology modified the miRAMM methodology to a MALDI-TOF MS (matrix-assisted laser desorption ionization time of flight MS) [63]. These techniques are more sensitive than protein electrophoresis or immunofixation at identifying the monoclonal light chain, albeit their sensitivity varies with the quantity of polyclonal background; in addition to identification, the size of this peak can be tracked quantitatively over time, providing a reproducible measure of disease burden with greater sensitivity than electrophoresis or immunofixation [64].

The utility of MALDI-TOF MS has been tested in a recent study [65] as a possible tool for assessing the treatment response. In a large series of patients with AL amyloidosis, they assessed the impact of FLC-MS negativity after treatment on overall survival and organ response rates. In a multivariate analysis, patients with no detectable residual monoclonal FLC by FLC-MS had significantly better OS and superior organ responses. FLC-MS negativity was an independent predictor of better survival in AL irrespective of the cardiac disease stage. This underlines the potential role of mass spectrometry not only in light chain identification, but also in the treatment response assessment and prognosis stratification.

Research on these methods is ongoing, and the IMWG Mass Spectrometry Committee, based on available evidence in 2021, issued recommendations regarding their use in plasma cell disorders [66]. They stated that mass spectrometry can be used as an alternative to immunofixation in clinical patient management but did not endorse its use as a replacement for the current immunochemical measurement of FLCs in amyloidosis and in other plasma cell disorders [66].

A new method is advancing for the detection of monoclonal proteins and response and serologic residual disease monitoring: the heavy/light chain (HLC) assay. This assay can quantify intact immunoglobulins in serum and has proven to be valuable in the diagnosis and monitoring of multiple myeloma. However, there is limited research on its application in AL amyloidosis. In a study published in 2025 [16], they evaluated the efficacy of HLC assay in AL amyloidosis patients at different disease stages, confirming its value as a complementary test in these patients. Compared with traditional methods, the HLC assay has unique advantages for discovering serum intact monoclonal immunoglobulins, including high efficiency, high automation detection and high sensitivity.

## 5. Purposes of Measuring Light-Chains in Tissues: Diagnosis, Risk Prediction

Accurate identification and characterization of amyloid deposits in tissues are essential for diagnosing AL amyloidosis. Although serum assays detecting monoclonal FLCs provide crucial initial diagnostic clues, definitive diagnosis requires demonstrating tissue deposition of these chains [67]. Typically, diagnosis involves biopsy of affected organs or surrogate sites such as abdominal fat or salivary glands, where amyloid deposits exhibit characteristic apple-green birefringence on Congo red staining under polarized light microscopy. Subsequent immunohistochemical staining for kappa or lambda light chains further establishes their clonality [68].

Mass spectrometry-based proteomic analyses have emerged as precise techniques for amyloid typing, allowing clear differentiation between AL amyloidosis and other types, guiding therapeutic decisions [69,70].

The biochemical and structural features of deposited light chains, including amino acid sequence variability and physicochemical properties, potentially predict disease severity and progression, especially regarding cardiac involvement [14,71]. Advances in proteomic and structural characterization of these deposits hold promise for refining risk stratification and individualizing therapy in AL amyloidosis patients.

## 6. Techniques for Tissue Analysis

The primary and most ancient method for identifying amyloid fibril deposits on formalin-fixed and paraffin-embedded (FFPE) tissue sections is Congo red staining, which dates back to 1922. Using a polarized light microscope, amyloid is shown as green birefringent deposits. Using an alkaline staining solution that is saturated with sodium chloride has further enhanced this technique. Tissue amyloid can also be found with metachromatic stains like crystal violet and fluorescent stains like thioflavin T or S [69].

Amyloid proteins on tissues are mostly identified and characterized by immunohistochemistry, either in electron microscopy (on glutaraldehyde-fixed tissue samples) using post-embedding immunogold techniques or in light microscopy (on FFPE tissues) using immunoperoxidase techniques [72]. This method is routinely available in most medical centers, offering a practical and cost-effective approach to amyloid typing [73]. However, in AL amyloidosis, its diagnostic accuracy is limited by variable sensitivity and specificity, with potential for false-positive or false-negative results [74], which often necessitates confirmation by mass spectrometry [73]. Despite the limitations (i.e., sensitivity) and potential drawbacks (see specificity) of antibody-based techniques, MS has been promoted as a viable means of identifying the type of amyloid fibrils. Four basic steps serve as the foundation for the two primary MS techniques: first, the sample’s proteins are all broken down, commonly using trypsin. The next step is to separate the pieces of 5–25 amino acids; liquid chromatography (LC) is the most often used method. The peptides are then ionized by subjecting the solution to high voltages. Tandem MS/MS analysis includes a mass/charge (*m*/*z*) measurement, peptide fragmentation upon contact with an inert gas, a technique known as collision induced fragmentation (CID), and a final MS measurement of the peptide’s unique CID, which enables the identification of the precise amino acid sequence. Using bioinformatic methods, the results are compared with reference databases in the last phase to ascertain the likelihood that each peptide comes from a particular protein [75].

A new technique is to use a fluorescence-equipped microscope to segment 10 µm tissue slices from FFPE specimens and isolate amyloid deposits using laser microdissection (LMD). Amyloid deposits may be effectively separated from the background with LMD, providing material for LC-MS/MS analysis and bioinformatic analysis (sensitivity and specificity of 98–100%) [75].

An alternative approach has been proposed [53], involving shotgun LC–MS/MS analysis applied to unfractionated tissue (typically fresh fat; FFPE samples can also be used) to generate comprehensive proteome maps that reveal the full set of proteins present in the sample [65]. Amyloid-positive samples are compared to negative control tissues, and amyloid identification is based on the alpha-value [68], a parameter quantifying the relative abundance of known amyloid proteins in patients versus controls.

The lack of knowledge regarding the fibrils’ spatial distribution is the main limit of both these latter options. The development of imaging-assisted MS was a recent solution to this problem. To distinguish ATTR from AL-lambda amyloidosis, matrix assisted laser desorption/ionization mass spectrometry imaging combined with ion mobility separation (MALDI-IMS MSI) has a 91% sensitivity and a 94% specificity [76].

When applying shotgun proteomic techniques to whole tissue, amyloidogenic fibrils are not directly identified; instead, their presence is inferred from co-deposited proteins that typically accompany them, such as Serum amyloid P, apolipoprotein IV, and apolipoprotein E [26].

Even if MS represents an excellent diagnostic tool for AL amyloidosis, its widespread implementation is limited by high costs and the requirement for specialized, multidisciplinary expertise. Furthermore, the reference databases necessary for amino acid sequence comparison and analysis remain under development (Figure 3). MS techniques can be applied to various tissue samples. In particular, fat aspirates, bone marrow, and biopsies from kidney, heart, or other affected organs have been shown to be suitable for MS analysis, enabling both amyloid typing and quantification of the pathogenic light chain clone [77].

Finally, while imaging techniques (echocardiography, cardiac magnetic resonance, nuclear imaging) do not measure circulating light chains directly, they provide valuable indirect insights into the pathogenic consequences of light chain excess. By characterizing the extent and pattern of organ involvement, especially in the heart, imaging complements biochemical assays and supports diagnosis, risk stratification, and monitoring of treatment response in AL amyloidosis.

In recent years, treatment of AL amyloidosis has evolved significantly. In addition to proteasome inhibitors and alkylating agents, the anti-CD38 monoclonal antibody daratumumab has demonstrated significant efficacy in improving hematologic response and organ function, particularly in patients ineligible for autologous stem cell transplantation, significantly improving the outcomes for patients with AL amyloidosis [78].

Currently, daratumumab in combination with cyclophosphamide, bortezomib, and dexamethasone (CyBorD or VCD) is the only first-line therapy approved for patients with AL amyloidosis [79].

**Figure 3 ijms-26-09511-f003:**
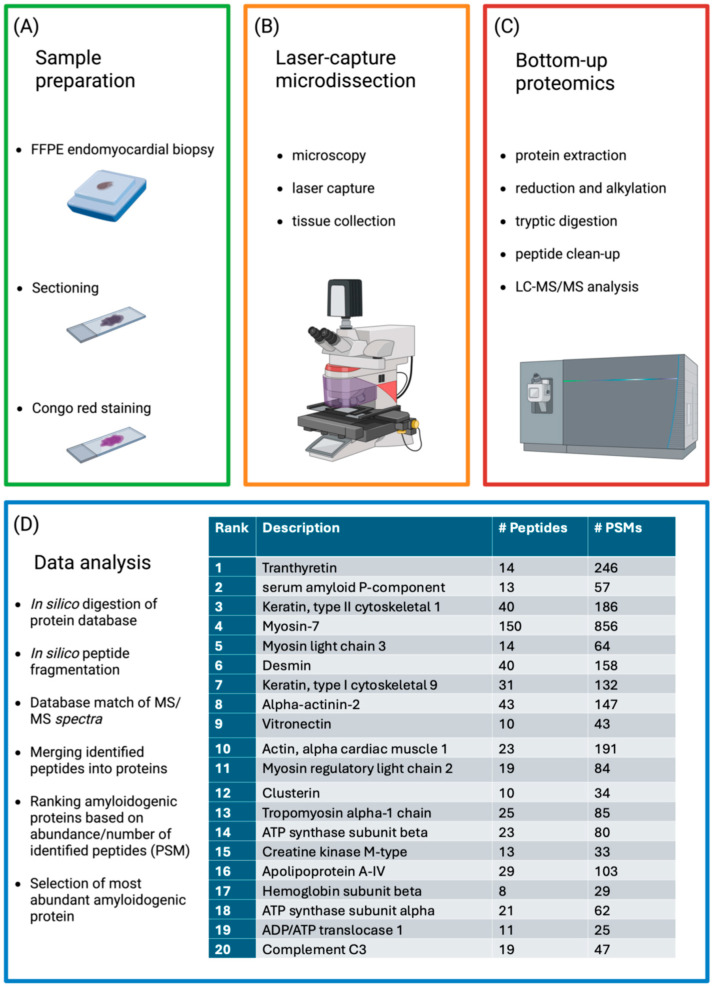
Proteomic workflow for amyloid typing from formalin-fixed, paraffin-embedded endomyocardial biopsy specimens. (**A**) A histological section is stained with Congo Red (without a coverslip) to identify amyloid deposits; (**B**) Congo Red-positive areas are isolated by laser microdissection and processed for proteomic analysis; (**C**) Fragmentation spectra are matched against an in silico-generated database; (**D**) The most abundant amyloidogenic protein is identified, while co-precipitating proteins represent part of a characteristic amyloid protein signature. FFPE, formalin-fixed, paraffin-embedded; LC–MS/MS, liquid chromatography coupled with tandem mass spectrometry; PSM, peptide-spectrum match. Modified with permission from Musetti et al., 2022 [80].

Emerging therapies targeting amyloid deposits or LC cytotoxicity, such as inhibitors of p38-MAPK or TFEB activators, may further address unmet needs in advanced disease [21].

## 7. Conclusions

Serum FLCs are crucial tools for identifying AL amyloidosis, determining the patient risk, and directing the treatment. There are five assays available. Depending on whether the FLCs are found in monomeric or dimeric forms, they may produce different outcomes. They are based on antigen–antibody recognition. The main drawbacks are that the two most commonly used assays yield differing results, and that even the Freelite^®^ assay might produce inconsistent outcomes depending on the platform and methodology employed. Western blot-based techniques, although not routinely applied in clinical practice, allow the differential analysis of monomeric and dimeric FLCs and may help clarify some of these discrepancies reported across immunoassays. A tissue biopsy is mandatory for the diagnosis of AL amyloidosis, and AL amyloid must be identified using immunohistochemistry and potentially proteomic techniques. Upon diagnosis of AL amyloidosis, precise risk stratification is essential for guiding treatment decisions and depends critically on circulating FLC levels. Lastly, circulating FLCs variations over time are significant markers of treatment response.

## Figures and Tables

**Figure 1 ijms-26-09511-f001:**
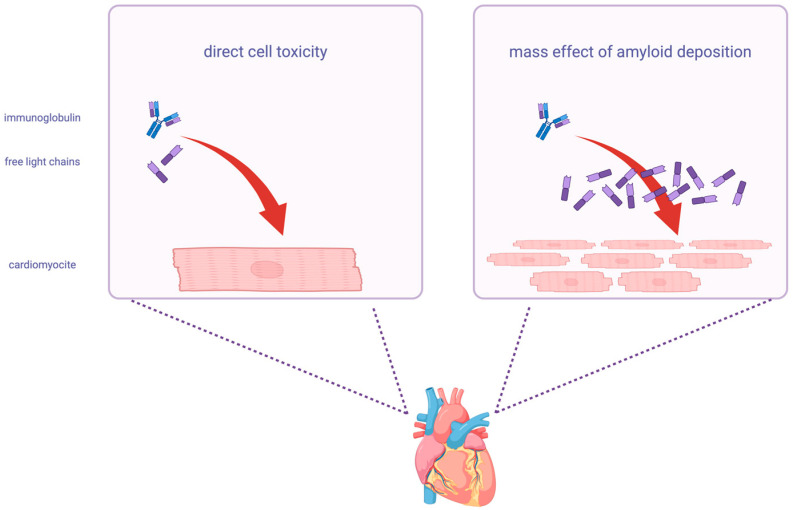
Mechanisms of damage for free light chains. On the left, the direct cytotoxic effect of light chains on cells; on the right, the mass effect of amyloid deposition on tissue. In the myocardial cell on the left box, there is a brief representation of the molecular cascade of the direct damage [23].

**Figure 2 ijms-26-09511-f002:**
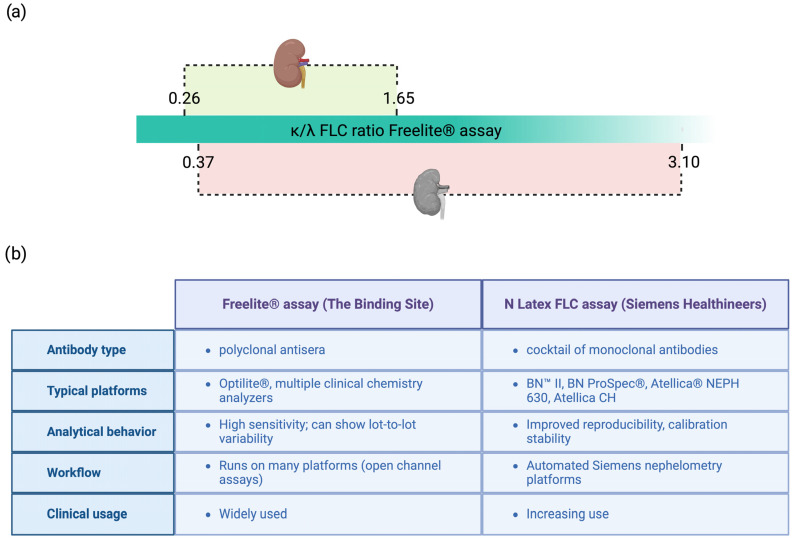
Normal range of kappa/lambda FLC ratio Freelite^®^ assay. The serum quantification of free light chains (FLCs) and the kappa/lambda ratio vary according to renal function. (**a**) A normal kappa/lambda ratio in case of normal renal function (0.26–1.65) has a 100% negative predictive value for light-chain cardiac amyloidosis (AL-CA). If a kidney disease is present, the normal range is 0.37–3.10. (**b**) Comparison of two widely used serum free light chain (sFLC) assays. Freelite^®^ (The Binding Site, Birmingham, UK) uses polyclonal antibodies and is available on multiple clinical chemistry analyzers. N Latex FLC (Siemens Healthineers, Erlangen, Germany) employs a monoclonal antibody mixture and is available exclusively on Siemens nephelometric platforms.

**Table 1 ijms-26-09511-t001:** Assays for free light-chain detection and quantification (ELISA, enzyme-linked immunosorbent assay; FLC, free light-chain; IMWG, International Myeloma Working Group).

Assay	Antisera	Method	Platform	Strengths	Possible Weaknesses	Quantification Potential
Freelite^®^ FLC	Polyclonal	- Nephelometric - Turbidimetric	- BN II System - Optilite	- Reference method for IMWG - First developed method - Long-standing on the market	Lot-to-lot and platform variability Underestimation due to antigen excess (hook effect) Poor linearity after dilution	Absolute quantification, but limited accuracy at very high concentrations
N Latex FLC	Monoclonal	Nephelometric	BN II System	High sensitivity	- Not interchangeable with Freelite - Limited detection of antigen excess	- Provides quantitative results; comparability across assays is restricted
Diazyme human kappa/lambda FLC	Polyclonal	Turbidimetric	Advia 1800	High sensitivity	- Underestimation due to antigen excess - Significant discrepancies at high FLC values	- Quantitative, but less reliable for lambda FLCs
Seralite^®^ FLC	Monoclonal	Lateral flow immunoassay	ADxRL5	- Reduced false negatives - Coupled measurement of kappa–lambda	- Lower clinical concordance (86–92%) - Higher variability in kappa/lambda ratio	- Quantitative, designed to minimize antigen excess, though ratios differ
Sebia FLC ELISA kappa/lambda	Polyclonal	ELISA sandwich	AP22 Elite	- Larger measurement interval - Reduced repetitions needed for FLC quantification	- Proportional positive bias at high concentrations	- Quantitative with improved linearity across broad ranges

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
