# Peer review of "Serum and Tissue Light-Chains as Disease Biomarkers in AL Amyloidosis"

_ijms, 2025, doi:10.3390/ijms26199511_

Round 1

Reviewer 1 Report

Comments and Suggestions for Authors

I thank the authors for their effort to review the role of light-chains in AL amyloidosis and the available methodologies for their detection both in serum and tissue. I have made few comments and suggestions to be considered in order to make your manuscript acceptable for publication in International Journal of Molecular Sciences.

Abstract:

  • Are we sure that AL amyloidosis is the most prevalent type of systemic amyloidosis? In my opinion, if properly diagnosed, wild type ATTR amyloidosis might even be more frequent.
  • Please, review the content of the first and second sentences of this section, as they seem somewhat repetitive.

Introduction:

  • Line 50: Again, I’m not sure that AL amyloidosis is the most prevalent type of systemic amyloidosis. Please, include a recent reference supporting this statement or do not include it in the manuscript.

Pathogenetic role of light-chains in AL amyloidosis: light chains as determinants of tissue damage

  • Please, consider to shorten the title as follows: “Pathogenetic role of light-chains in AL amyloidosis”

Scopes of measuring circulating light-chains: diagnosis, risk prediction, assessment of response to therapy: light chains as diagnostic and prognostic tools

  • Again, please consider shortening the title, and notice that you’ve written a colon twice in one sentence. I suggest as title of this section: “The role of monoclonal light-chains in diagnosis, risk prediction and response assessment”.
  • Line 135: I think the statement “with FLCs detected in only approximately half of cases” is confusing. In my experience, immunofixation allows the identification of the monoclonal protein (including the light chain type) in more than 90% of cases, while it is quantifiable by ELP in only half of them.
  • Figure 2: The image of the kidney representing the situation of “kidney disease” is not clear to me.

Methods to measure circulating light-chains

  • Table 1: What does “v” mean? (title for fifth column of the table).
  • Line 178: Should there be a period between “ … International Myeloma Working Group.” and “The Freelite® assay …”?
  • Line 182: Kappa and lambda light chains instead of just chains (here and throughout the manuscript).
  • Lines 261-263: I think the reader would appreciate having a clearer understanding of the potential of these techniques to not only identify but quantify de FLC.

Purposes of measuring light-chains in tissues: diagnosis, risk prediction.

  • Please, delete the stop at the end of the title.

Techniques for tissue analysis

  • The last sentence of the first paragraph and the first sentence of the second paragraph seem to refer to the same thing (duplication?).
  • Line 334: Please, include in the discussion (in this section) which samples would be suitable for tissue analysis using MS techniques (according to available data).
  • Line 358: As stated in reference 60, “daratumumab in combination with cyclophosphamide, bortezomib and dexamethasone (CyBorD or VCD) is the only first-line therapy approved for patients with AL amyloidosis”, but not only in those who are not candidates for ASCT.

Conclusions

  • No comments.

Author Response

Response to Reviewers

Reviewer 1

I thank the authors for their effort to review the role of light-chains in AL amyloidosis and the available methodologies for their detection both in serum and tissue. I have made few comments and suggestions to be considered in order to make your manuscript acceptable for publication in International Journal of Molecular Sciences.

Abstract:

  • Are we sure that AL amyloidosis is the most prevalent type of systemic amyloidosis? In my opinion, if properly diagnosed, wild type ATTR amyloidosis might even be more frequent.

We thank the Reviewer for the suggestion. According to current epidemiological data, AL amyloidosis remains the most prevalent form of systemic amyloidosis in clinical practice, particularly in Western countries.

For example:

  • In a large UK cohort, AL accounted for 65% of systemic amyloidosis diagnoses, compared to 7% for wild-type ATTR and 10% for hereditary ATTR (Wechalekar et al., 2016, Br J Haematol).
  • A systematic review by Merlini and colleagues confirmed an annual incidence of approximately 12 cases per million for AL in the USA/EU, with an estimated prevalence of 30,000–45,000 cases, making it the most frequently diagnosed type in clinical settings (Merlini et al., 2020, JAMA).

However, wild-type ATTR amyloidosis is substantially underdiagnosed. Data from autopsy studies and targeted screening in high-risk groups (e.g., HFpEF, severe aortic stenosis) suggest that ATTR-wt may in fact be more common in the general population than AL. Yet, because it is often clinically silent or misattributed, its recorded prevalence in clinical practice remains lower. In other words, while ATTR-wt could be the most frequent type if systematically identified, current real-world diagnostic data still place AL at the top.

  • Please, review the content of the first and second sentences of this section, as they seem somewhat repetitive.

We thank the Reviewer for the note. We have eliminated the repetition that was present in the second sentence: “AL amyloidosis is a protein misfolding disorder characterised by the tissue deposition of monoclonal light-chains (LCs) produced by neoplastic plasma cells”.

Introduction:

  • Line 50: Again, I’m not sure that AL amyloidosis is the most prevalent type of systemic amyloidosis. Please, include a recent reference supporting this statement or do not include it in the manuscript.

We thank the Reviewer for the note. We added the references in order to support this statement.

Pathogenetic role of light-chains in AL amyloidosis: light chains as determinants of tissue damage:

  • Please, consider to shorten the title as follows: “Pathogenetic role of light-chains in AL amyloidosis”

We agree with the Reviewer. The text was modified as follows: “Pathogenetic role of light-chains in AL amyloidosis”

Scopes of measuring circulating light-chains: diagnosis, risk prediction, assessment of response to therapy: light chains as diagnostic and prognostic tools:

  • Again, please consider shortening the title, and notice that you’ve written a colon twice in one sentence. I suggest as title of this section: “The role of monoclonal light-chains in diagnosis, risk prediction and response assessment”.

We thank the Reviewer for the note, and we are sorry for the grammatical mistake. We have corrected title of the section, as suggested, with the following one: “The role of monoclonal light-chains in diagnosis, risk prediction and response assessment”

  • Line 135: I think the statement “with FLCs detected in only approximately half of cases” is confusing. In my experience, immunofixation allows the identification of the monoclonal protein (including the light chain type) in more than 90% of cases, while it is quantifiable by ELP in only half of them.

We thank the Reviewer for the notification. We have corrected the sentence and added supporting evidence. The new sentence is the following one: “AL amyloidosis is characterized by relatively low concentrations of circulating monoclonal proteins. These proteins can be identified in approximately 90–95% of patients using combined serum and urine immunofixation (Merlini & Palladini, Blood, 2008). In contrast, serum protein electrophoresis has a considerably lower sensitivity, detecting the monoclonal component in only about half of cases (Bochtler et al., Haematologica, 2008).

  • Figure 2: The image of the kidney representing the situation of “kidney disease” is not clear to me.

We thank the Reviewer for the punctualization. We modified the image to enhance the difference between the scenario of “healthy kidney” and “kidney disease”.

Methods to measure circulating light-chains:

  • Table 1: What does “v” mean? (title for fifth column of the table).

We are sorry for the typing mistake. We have completed the column adding “strenghts” instead of “v”.

  • Line 178: Should there be a period between “ … International Myeloma Working Group.” and “The Freelite® assay …”?

We thank the Reviewer for the notification. The first sentence that end with … International Myeloma Working Group.” is part of the Table 1 caption where the acronyms are explained. The following sentence, that begins with “The Freelite® assay …”  is instead the beginning of the next paragraph. We have corrected the period adding the explanation of the acronyms in the table caption, in order to avoid misunderstandings.

  • Line 182: Kappa and lambda lightchains instead of just chains (here and throughout the manuscript).

We thank the Reviewer for the note. We have corrected the whole manuscript adding “light” to the kappa and lambda chains where it was not specified.

  • Lines 261-263: I think the reader would appreciate having a clearer understanding of the potential of these techniques to not only identify but quantify de FLC.

We thank the Reviewer for the request of clarification. We modified the entire paragrapher as follows: “Five commercial diagnostic assays are currently available for measuring serum FLC: Freelite®, N Latex FLC, Diazyme, Seralite®, and Sebia FLC assays. Each method is based on antibodies able to identify specific epitopes of the FLC. These are described as “hidden” because they seem to be visible only when the LCs are actually free, while they cannot be recognised when they are paired with heavy chains. Table 1 summarises the main characteristics of these assays. The Freelite® assay (The Binding Site Group Ltd, Birmingham, UK) was the first method available on automated instruments and remained the only option for a decade. It is considered the reference assay in the International Myeloma Working Group (IMWG) recommendations [32], [33]. Polyclonal antibodies against kappa and lambda light chains are obtained by immunizing sheep with urinary light chains from patients with monoclonal proteinuria (Bence–Jones proteins). These antibodies are subsequently purified through affinity techniques, in which they are exposed to intact immunoglobulins, and only those showing specific reactivity are retained. They are coated onto polystyrene latex particles and measured by nephelometry or turbidimetry. Calibration uses human serum with defined FLC content, and reference intervals are provided for the BN II System (Siemens Healthineers Diagnostics GmbH, Marburg, Germania, nephelometric technique) [33]. Limitations include variability across different instruments even when the same reagents are used, as well as underestimation due to antigen excess. Thus, while the Freelite® assay enables absolute quantification, its accuracy is reduced at very high concentrations or after dilution [34], [35].[35]

The N Latex FLC assays (Siemens Healthineers Diagnostics GmbH, Marburg, Germany) are latex-based nephelometric assays designed specifically for the Siemens BN II systems. They employ murine monoclonal antibodies recognizing epitopes in the constant domain of FLCs, with detergents added to reduce nonspecific binding. Calibration is performed with purified polyclonal FLCs in buffered solution containing human albumin, and values are standardized against Freelite®. In AL amyloidosis this approach is considered valid, as quantification levels between the assays are highly concordant [14]. In contrast, discrepant results have occasionally been reported in multiple myeloma [36]. Comparative studies have shown that while the two assays demonstrate similar clinical utility, they are not analytically equivalent and therefore should not be used interchangeably for patient monitoring [37]. Accordingly, N Latex assays provide quantitative results, yet cross-platform equivalence remains an issue.

The Diazyme FLC assays (Diazyme Laboratories Inc., Poway, CA, USA) use latex-enhanced immunoturbidimetry with rabbit polyclonal antisera. Two comparative studies against Freelite® reported acceptable agreement for kappa light chains but marked discrepancies for lambda light chains [38], [39]. So, although the method yields quantitative values, reliability is reduced for lambda measurements, particularly at higher concentrations

The Seralite® assay (Abingdon Health/Sebia, Evry, France) enables simultaneous determination of kappa and lambda FLCs using a competitive inhibition mechanism. Each test strip incorporates immobilized FLCs that compete with patient FLCs for binding to gold nanoparticle-conjugated monoclonal antibodies. The signal, read by the ADxLR5 device, is inversely proportional to analyte concentration [40]. This design reduces the risk of antigen excess effects. No significant differences are found in median FLC values compared to the Freelite® test, although the kappa/lambda ratio of Seralite® has a wider range [40]. Hence, Seralite® is quantitative, with better performance in high-concentration settings, but interpretation of ratios may differ.

The Sebia enzyme-linked immunosorbent assay (ELISA) (Sebia, Evry, France) uses a sandwich format with rabbit polyclonal antibodies, with the secondary antibody labeled by horseradish peroxidase [41]. It has been fully automated on the AP22 ELITE processor. The assay correlates well with Freelite® and, due to its broader dynamic range, reduces the need for repeat dilutions. However, biases emerge at higher concentrations [42]. This approach therefore provides robust quantitative measurement with improved linearity compared to other immunoassays.

Overall, serum FLC measurement significantly increases sensitivity in screening panels for AL amyloidosis, leading to its inclusion in IMWG recommendations since 2009 [43], [44].

Despite improvements, immunoassays remain prone to technical limitations such as the hook effect. Alternative technologies are being explored to improve precision. Mass spectrometry (MS) has become particularly attractive with advances in accessibility and instrumentation. By targeting the unique amino acid sequence in the complementarity-determining region (CDR), MS can provide both identification and accurate quantification of monoclonal light chains.

The clonotypic peptide approach requires prior sequencing of the clonal light chain to monitor unique peptides by LC–MS/MS [45], [46]. Although technically demanding, once established it offers highly sensitive and accurate quantification, particularly for measurable residual disease. By contrast, the miRAMM (monoclonal immunoglobulin rapid accurate mass measurement) technique does not require prior sequence knowledge [47]. Immunoglobulins were first enriched from samples, then reduced to release free light chains, which were subsequently subjected to LC-ESI-TOF MS analysis. The monoclonal light chain appeared as a single peak when the mass spectra of multiply charged light chains were deconvoluted to determine their molecular mass. The distinct m/z distribution of their constant sections allowed for the differentiation of kappa and lambda light chains. Without requiring chromatography, the same team that created this initial technology modified the miRAMM methodology to a MALDI-TOF MS (matrix-assisted laser desorption ionization time of flight MS) [48]. These techniques are more sensitive than protein electrophoresis or immunofixation at identifying the monoclonal light chain, albeit their sensitivity varies with the quantity of polyclonal background; in addition to identification, the size of this peak can be tracked quantitatively over time, providing a reproducible measure of disease burden with greater sensitivity than electrophoresis or immunofixation  [49].

Research on these methods is ongoing, and the IMWG Mass Spectrometry Committee, based on available evidence in 2021, issued recommendations regarding their use in plasma cell disorders [50]. They stated that mass spectrometry can be used as an alternative to immunofixation in clinical patient management but did not endorse its use as a replacement for the current immunochemical measurement of FLCs in amyloidosis and in other plasma cell disorders”.

Then we added the last column to the Table 1 to underline the aspects required.

Assay

Antisera

Method

Platform

Strengths

Weaknesses

Quantification potential

Freelite® FLC

Polyclonal

Nephelometric / Turbidimetric

BN II, Optilite

Reference method for IMWG; longest clinical use

Lot-to-lot and platform variability; hook effect; poor linearity after dilution

Absolute quantification, but limited accuracy at very high concentrations

N Latex FLC

Monoclonal

Nephelometric

BN II

High sensitivity

Not interchangeable with Freelite®; limited antigen excess detection

Provides quantitative results; comparability across assays is restricted

Diazyme FLC

Polyclonal

Turbidimetric

Advia 1800

High sensitivity

Antigen excess effect; discrepancies at high FLC levels, especially lambda

Quantitative, but less reliable for lambda FLCs

Seralite® FLC

Monoclonal

Lateral flow immunoassay

ADxRL5

Reduced false negatives; simultaneous kappa–lambda

Lower concordance (86–92%); variability in ratios

Quantitative, designed to minimize antigen excess, though ratios differ

Sebia ELISA FLC

Polyclonal

ELISA sandwich

AP22 ELITE

Wide measurement interval; fewer dilutions required

Positive bias at high levels

Quantitative with improved linearity across broad ranges

Purposes of measuring light-chains in tissues: diagnosis, risk prediction:

  • Please, delete the stop at the end of the title.

We apologize for the mistake. We have eliminated the stop at the end of the title.

Techniques for tissue analysis

  • The last sentence of the first paragraph and the first sentence of the second paragraph seem to refer to the same thing (duplication?).

We thank the Reviewer for the observation. We have eliminated the repetition that was present in this section: “Amyloid proteins are identified and classified in tissue sections using specific antibodies directed against kappa or lambda Ig light-chains, combined with immunoperoxidase or indirect immunofluorescence techniques”.

  • Line 334: Please, include in the discussion (in this section) which samples would be suitable for tissue analysis using MS techniques (according to available data).

We followed the suggestion integrating the paragraph with some examples of sites for biopsy: “Even if MS represents an excellent diagnostic tool for AL amyloidosis, its widespread implementation is limited by high costs and the requirement for specialized, multidisciplinary expertise. Furthermore, the reference databases necessary for amino acid sequence comparison and analysis remain under development (Figure 3). MS techniques can be applied to various tissue samples. In particular, fat aspirates, bone marrow, and biopsies from kidney, heart, or other affected organs have been shown to be suitable for MS analysis, enabling both amyloid typing and quantification of the pathogenic light chain clone (doi 10.1016/j.mayocp.2020.06.029)”

  • Line 358: As stated in reference 60, “daratumumab in combination with cyclophosphamide, bortezomib and dexamethasone (CyBorD or VCD) is the only first-line therapy approved for patients with AL amyloidosis”, but not only in those who are not candidates for ASCT.

We thank the Reviewer for the punctualizaiton. We have deleted the last part of the sentence. Now it figures “Currently, daratumumab in combination with cyclophosphamide, bortezomib, and dexamethasone (CyBorD or VCD) is the only first-line therapy approved for patients with AL amyloidosis”

Reviewer 2 Report

Comments and Suggestions for Authors

This manuscript deals with an important diagnostic issue of AL amyloidosis  - identification of light chain amyloid deposits in tissue and detection of amyloidogenic FLC in serum. 

Below are some comments and suggestions to improve this manuscript.

  1. Introduction: authors should present a more detailed description of structural features of light chain fibrillar deposits. This should also include explanation regarding truncated light chains often found in AL deposits. Presence of amyloid-associated proteins should be also mentioned here.
  2. Lane 87 (in regard to the issue of germline gene usage in AL): authors should take into consideration more recent publications on this matter, such as Morgan et al., 2025.
  3. Lanes 145-162, Freelite against N Latex assay:  Authors should also mention the existing differences between the calibrators in these two assays (Kaplan and Jacobs, 2021).
  4. Lanes 298-301: Immunohistochemistry is a routine technique used in most medical centers. Advantages and limitations of this antibody-based approach in AL diagnosis should be specified in more details.
  5. Lanes 318-322 and 328–330: The statements made in these two paragraphs seem to be contradictive. Please clarify.
  6. Conclusions: “Five assays are available. They are based on antigen-antibody  recognition and produce different results possibly depending on whether the FLCs are found in monomeric or dimeric forms”. However, in this review, authors do not present and discuss the publications of Western blot-based techniques which do allow differential analysis of monomeric and dimeric FLC. 

Author Response

Reviewer 2

This manuscript deals with an important diagnostic issue of AL amyloidosis - identification of light chain amyloid deposits in tissue and detection of amyloidogenic FLC in serum. 

Below are some comments and suggestions to improve this manuscript.

  • Introduction: authors should present a more detailed description of structural features of light chain fibrillar deposits. This should also include explanation regarding truncated light chains often found in AL deposits. Presence of amyloid-associated proteins should be also mentioned here.

We thank the reviewer for this insightful comment. We agree that a more detailed description of the structural features of light chain fibrillar deposits, including truncated light chains and amyloid-associated proteins, would strengthen the introduction. Accordingly, we have revised the introduction to incorporate these points:

  • Lane 87 (in regard to the issue of germline gene usage in AL): authors should take into consideration more recent publications on this matter, such as Morgan et al., 2025.

We thank the Reviewer for the suggestion. We have added the publication in the references and we have updated the manuscript.

  • Lanes 145-162, Freelite against N Latex assay: Authors should also mention the existing differences between the calibrators in these two assays (Kaplan and Jacobs, 2021).

We thank the Reviewer for the note. We have added information about the differences between the calibrators in the Freelite and N Latex assay. We have updated the references as suggested with the manuscript of Kaplan and Jacobs.

  • Lanes 298-301: Immunohistochemistry is a routine technique used in most medical centers. Advantages and limitations of this antibody-based approach in AL diagnosis should be specified in more details.

We thank the reviewer for this constructive comment. Accordingly, we have expanded the section on immunohistochemistry to provide more details regarding its routine application, strengths, and drawbacks. So we added this part: “This method is routinely available in most medical centers, offering a practical and cost-effective approach to amyloid typing. However, in AL amyloidosis, its diagnostic accuracy is limited by variable sensitivity and specificity, with potential for false-positive or false-negative results, which often necessitates confirmation by mass spectrometry.”

  • Lanes 318-322 and 328–330: The statements made in these two paragraphs seem to be contradictive. Please clarify.

We thank the Reviewer for this helpful observation. The two statements are not contradictory but complementary: the first one describes the technical approach and its ability to generate comprehensive proteome maps from unfractionated tissue, while the second clarifies the interpretative limitation of that approach with respect to direct detection of amyloid fibrils. To remove any possible confusion we have revised both sentences as follows.

Lines 318-322. Original: “An alternative approach has been proposed [52], involving shotgun LC-MS/MS analysis to generate proteome maps of unfractionated tissue (typically fresh fat, although FFPE samples can also be used).”

Revised: “An alternative approach has been proposed [52], involving shotgun LC–MS/MS analysis applied to unfractionated tissue (typically fresh fat; FFPE samples can also be used) to generate comprehensive proteome maps that reveal the full set of proteins present in the sample.”

Lines 328-330.  Original: “Using shotgun proteomic techniques to analyze the entire tissue, amyloidogenic fibrils can be identified only by the presence of specific proteins that often deposit with them: Serum amyloid P, apolipoprotein IV, and apolipoprotein E.”

Revised: “When applying shotgun proteomic techniques to whole tissue, amyloidogenic fibrils are not directly identified; instead, their presence is inferred from co-deposited proteins that typically accompany them, such as Serum amyloid P, apolipoprotein IV, and apolipoprotein E.”

We trust that these clarifications remove the apparent discrepancy by explicitly stating the technical scope of the shotgun LC–MS/MS approach and noting its interpretative limitation regarding direct identification of amyloid fibrils.

Conclusions: “Five assays are available. They are based on antigen-antibody  recognition and produce different results possibly depending on whether the FLCs are found in monomeric or dimeric forms”. However, in this review, authors do not present and discuss the publications of Western blot-based techniques which do allow differential analysis of monomeric and dimeric FLC. 

We thank the reviewer for this valuable comment. We agree that Western blot-based approaches, which can differentiate monomeric from dimeric free light chains, are relevant to the discussion. In line with this suggestion, we have revised the Conclusions section to include a brief reference to these techniques and their potential contribution. In particular, we added this sentence in the Conclusions section: “Western blot-based techniques, although not routinely applied in clinical practice, allow the differential analysis of monomeric and dimeric FLCs and may help clarify some of these discrepancies reported across immunoassays.”.

Reviewer 3 Report

Comments and Suggestions for Authors

The authors summarized serum and tissue light-chains as biomarkers of AL amyloidosis. The pathogenesis of AL amyloidosis, measuring methods of serum free light chain and the analysis technique for tissue specimens are described. This review is well written. I have just a trivial comment. In the title, “targets for the treatment in AL amyloidosis” are included, but your contents rarely describe the treatment, such as CD38 antibody, and other new treatment targets. The title is overly expressive.

Author Response

Reviewer 3

The authors summarized serum and tissue light-chains as biomarkers of AL amyloidosis. The pathogenesis of AL amyloidosis, measuring methods of serum free light chain and the analysis technique for tissue specimens are described. This review is well written. I have just a trivial comment. In the title, “targets for the treatment in AL amyloidosis” are included, but your contents rarely describe the treatment, such as CD38 antibody, and other new treatment targets. The title is overly expressive.

We thank the Reviewer for the comment and we agree with His/Her consideration. The title is overly expressive for the contents of the manuscript. We have eliminated from the title the part “targets for the treatment”. The new title will be “Serum and tissue light-chains as disease biomarkers in AL amyloidosis”.

Reviewer 4 Report

Comments and Suggestions for Authors

The manuscript presents a review of the role of light chains (LCs) in AL amyloidosis, focusing on both their pathogenic role and their use as diagnostic/prognostic biomarkers. While the topic is clinically relevant, the review lacks sufficient novelty. Much of the presented material closely follows content already published in prior literature, including the 2022 article in Vessel Plus (Camerini L, Aimo A, Pucci A, Castiglione V, Musetti V, Masotti S, Caponi L, Vergaro G, Passino C, Clerico A, Franzini M, Emdin M. Serum and tissue light-chains as disease biomarkers and targets for treatment in AL amyloidosis. Vessel Plus. 2022;6:59. http://dx.doi.org/10.20517/2574-1209.2022.06), to which the authors have not provided appropriate attribution. Several sections are essentially reformulations of existing review material without adding substantial new insights, hypotheses, or critical interpretation. This raises concerns about both originality and scholarly rigor. Below are some additional comments that also prevent the manuscript from being accepted for publication. However, the main issue, as noted above, is that much of the review material has already been published in 2022.

Comments:

Lines 20-26. This text corresponds to the abstract of the article by Camerini L, Aimo A, Pucci A, Castiglione V, Musetti V, Masotti S, Caponi L, Vergaro G, Passino C, Clerico A, Franzini M, Emdin M. Serum and tissue light-chains as disease biomarkers and targets for treatment in AL amyloidosis. Vessel Plus. 2022;6:59. http://dx.doi.org/10.20517/2574-1209.2022.06

Lines 52-53, 56-58, 92-101, 107-118, 129-160, 170-183, 193-204, 211-269, 288-330, 363-373. This text corresponds to the published text in the article by Camerini L, Aimo A, Pucci A, Castiglione V, Musetti V, Masotti S, Caponi L, Vergaro G, Passino C, Clerico A, Franzini M, Emdin M. Serum and tissue light-chains as disease biomarkers and targets for treatment in AL amyloidosis. Vessel Plus. 2022;6:59. http://dx.doi.org/10.20517/2574-1209.2022.06

The topic of the review is relevant, but the number of literature references (60) is relatively small for a literature review.

Figure 3 is presented at insufficient resolution, making the text difficult to read.

Author Response

Reviewer 4

The manuscript presents a review of the role of light chains (LCs) in AL amyloidosis, focusing on both their pathogenic role and their use as diagnostic/prognostic biomarkers. While the topic is clinically relevant, the review lacks sufficient novelty. Much of the presented material closely follows content already published in prior literature, including the 2022 article in Vessel Plus (Camerini L, Aimo A, Pucci A, Castiglione V, Musetti V, Masotti S, Caponi L, Vergaro G, Passino C, Clerico A, Franzini M, Emdin M. Serum and tissue light-chains as disease biomarkers and targets for treatment in AL amyloidosis. Vessel Plus. 2022;6:59. http://dx.doi.org/10.20517/2574-1209.2022.06), to which the authors have not provided appropriate attribution. Several sections are essentially reformulations of existing review material without adding substantial new insights, hypotheses, or critical interpretation. This raises concerns about both originality and scholarly rigor. Below are some additional comments that also prevent the manuscript from being accepted for publication. However, the main issue, as noted above, is that much of the review material has already been published in 2022.

Comments:

  • Lines 20-26. This text corresponds to the abstract of the article by Camerini L, Aimo A, Pucci A, Castiglione V, Musetti V, Masotti S, Caponi L, Vergaro G, Passino C, Clerico A, Franzini M, Emdin M. Serum and tissue light-chains as disease biomarkers and targets for treatment in AL amyloidosis. Vessel Plus. 2022;6:59. http://dx.doi.org/10.20517/2574-1209.2022.06

We thank the reviewer for the note. We have modified the text between the lines 20-26. The new text will figure like this: “Circulating free LC (FLC) measurement, which can be done by mass spectrometry or antibody-based techniques, is a crucial tool for AL amyloidosis diagnosis, risk assessment, and management. Additionally, diagnosing AL amyloidosis requires accurate detection of LC deposits in tissues”.

  • Lines 52-53, 56-58, 92-101, 107-118, 129-160, 170-183, 193-204, 211-269, 288-330, 363-373. This text corresponds to the published text in the article by Camerini L, Aimo A, Pucci A, Castiglione V, Musetti V, Masotti S, Caponi L, Vergaro G, Passino C, Clerico A, Franzini M, Emdin M. Serum and tissue light-chains as disease biomarkers and targets for treatment in AL amyloidosis. Vessel Plus. 2022;6:59. http://dx.doi.org/10.20517/2574-1209.2022.06

We thank the Reviewer for the consideration and we are sorry for the repetitions. We have corrected all the lines that the Reviewer highlighted. The new lines will figure like these:

Lines 52-53: “The most prevalent  type, especially in Western countries (Wechalekar et al., 2016, Br J Haematol) (Merlini et al., 2020, JAMA), is amyloid light-chain (AL) amyloidosis, which is defined by the accumulation of free immunoglobulin light chains (FLCs) of antibodies in the tissue as a result of a plasma cell dysfunction. There are two types of clonal light-chains (LCs): lambda (found in 70–80% of individuals) and kappa (20–30%)”.

Lines 56-58: In the United States, around 4,000 persons receive an AL amyloidosis diagnosis annually, with the majority of these diagnoses occurring in those between the ages of 50 and 65 [7]. Prior to 2010, the prevalence was between 8.8 and 15.5 cases per million; after 2010, it was between 40 and 58 cases per million [8].

Lines 92-101: Direct cell toxicity is thought to be the primary mechanism of injury exerted by amyloidogenic precursors [11]. One possible explanation for the quicker clinical development seen in AL-CA as opposed to ATTR-CA is direct injury to cardiomyocytes [12]. The mass effect of amyloid deposition, which alters tissue architecture and jeopardizes organ function, is a representation of the second damage process (Figure 1). AL amyloidosis has been referred to as a "toxic infiltrative cardiomyopathy" because of this. Cells that absorb soluble LCs are inherently poisonous[13], [14]. Apoptosis, oxidative stress, mitochondrial dysfunction, poor calcium handling and contractility, aberrant autophagy, and lysosomal dysfunction are among the alterations that have been seen in animal models of light-chain cardiotoxicity [15], [16].

Lines 107-118: The detrimental effects of amyloidogenic light chains have been investigated in the two predominant types of heart cells: cardiomyocytes [17] and fibroblasts [18]. Studies demonstrated that amyloidogenic LCs interact with proteins and other biological components, possibly by causing them to malfunction or sequester. In particular, amyloidogenic LCs seem to interfere with important metabolic pathways in mitochondria by interacting with peroxisomal ACOX1 (i.e., the first enzyme of the fatty acid beta-oxidation pathway) and OPA1 (optic atrophy 1-like protein), which is found in the inner mitochondrial membrane [19], thereby impairing key metabolic pathways. Furthermore, the pathophysiology and organ tropism of amyloidosis seem to be significantly influenced by the extracellular matrix (ECM) [20]. Exposure of cells to amyloidogenic proteins induces alterations in the secretion and processing of ECM components, with fibril deposition occurring in tight spatial association with collagen and glycosaminoglycans.

Lines 129-160: Interestingly, lambda FLCs are primarily found as dimers, while kappa FLCs are primarily found as monomers. Serum immunofixation (S-IF) can be used to further characterize monoclonal serum immunoglobulins, which may appear as an abnormal peak on serum protein electrophoresis (ELP). On the other hand, when only an excessive amount of monoclonal FLCs are produced, the ELP pattern typically remains unaltered [23]. AL amyloidosis is distinguished by comparatively modest levels of circulating monoclonal proteins in contrast to other plasma cell diseases. These proteins can be identified in approximately 90–95% of patients using combined serum and urine immunofixation (Merlini & Palladini, Blood, 2008). In contrast, serum protein electrophoresis has a considerably lower sensitivity, detecting the monoclonal component in only about half of cases (Bochtler et al., Haematologica, 2008).

The advent of automated serum FLC assays has markedly improved the quantification of serum FLC levels, the kappa/lambda ratio and the ratio between involved and uninvolved FLC (i/uFLC), thereby revolutionizing the diagnosis and monitoring of plasma cell disorders [24]. In subjects with normal kidney function, the kappa/lambda ratio ranges between 0.26 and 1.65 when using the Freelite® assay. In patients with chronic kidney disease, the reticuloendothelial system becomes more important for FLC removal, and the ratio can increase up to 3 with the same assay (range 0.37-3.10) (Figure 2) [25]. In patients with cardiac disease, a normal kappa/lamba FLC ratio has a 100% negative predictive value for AL-CA [26].

FLC measurement is also a crucial parameter for the follow-up of individuals already diagnosed with the disease. In the study of amyloidosis, there has been a growing interest in the difference between the concentration of uninvolved FLCs (dFLC) and involved FLCs, which is caused by clonal growth. One of the best indicators of overall survival in AL amyloidosis is dFLC (as determined by the Freelite® assay); the prognostic MAYO2012 score includes the 180 mg/L cut-off [27]. Furthermore, a >50% decrease in dFLC or a reduction to <40 mg/L are important criteria of response to treatment [28]. A later comparison study found that using N Latex FLC to assess FLCs resulted in a lower predictive cut-off for dFLC (165 mg/L) [14]. Even though the N Latex assays have similar diagnostic sensitivity and prognostic value, they cannot be utilized for disease staging or evaluating therapy response since studies that incorporate dFLC into staging systems and response criteria are based on the Freelite® assay. The oligomeric form of FLCs has also been linked to differences between these two approaches [29], [30]. Indeed, it has recently been shown that the N Latex reagent strongly recognizes the lambda FLC monomers, while the Freelite® test better detects lambda FLC dimers connected by an inter-chain disulphide bridge [31].

Lines 170-183, Lines 193-204, Lines 211-269: Five commercial diagnostic assays are currently available for measuring serum FLC: Freelite®, N Latex FLC, Diazyme, Seralite®, and Sebia FLC assays. Each method is based on antibodies able to identify specific epitopes of the FLC. These are described as “hidden” because they seem to be visible only when the LCs are actually free, while they cannot be recognised when they are paired with heavy chains. Table 1 summarises the main characteristics of these assays. The Freelite® assay (The Binding Site Group Ltd, Birmingham, UK) was the first method available on automated instruments and remained the only option for a decade. It is considered the reference assay in the International Myeloma Working Group (IMWG) recommendations [32], [33]. Polyclonal antibodies against kappa and lambda light chains are obtained by immunizing sheep with urinary light chains from patients with monoclonal proteinuria (Bence–Jones proteins). These antibodies are subsequently purified through affinity techniques, in which they are exposed to intact immunoglobulins, and only those showing specific reactivity are retained. They are coated onto polystyrene latex particles and measured by nephelometry or turbidimetry. Calibration uses human serum with defined FLC content, and reference intervals are provided for the BN II System (Siemens Healthineers Diagnostics GmbH, Marburg, Germania, nephelometric technique) [33]. Limitations include variability across different instruments even when the same reagents are used, as well as underestimation due to antigen excess. Thus, while the Freelite® assay enables absolute quantification, its accuracy is reduced at very high concentrations or after dilution [34], [35].[35]

The N Latex FLC assays (Siemens Healthineers Diagnostics GmbH, Marburg, Germany) are latex-based nephelometric assays designed specifically for the Siemens BN II systems. They employ murine monoclonal antibodies recognizing epitopes in the constant domain of FLCs, with detergents added to reduce nonspecific binding. Calibration is performed with purified polyclonal FLCs in buffered solution containing human albumin, and values are standardized against Freelite®. In AL amyloidosis this approach is considered valid, as quantification levels between the assays are highly concordant [14]. In contrast, discrepant results have occasionally been reported in multiple myeloma [36]. Comparative studies have shown that while the two assays demonstrate similar clinical utility, they are not analytically equivalent and therefore should not be used interchangeably for patient monitoring [37]. Accordingly, N Latex assays provide quantitative results, yet cross-platform equivalence remains an issue.

The Diazyme FLC assays (Diazyme Laboratories Inc., Poway, CA, USA) use latex-enhanced immunoturbidimetry with rabbit polyclonal antisera. Two comparative studies against Freelite® reported acceptable agreement for kappa light chains but marked discrepancies for lambda light chains [38], [39]. So, although the method yields quantitative values, reliability is reduced for lambda measurements, particularly at higher concentrations

The Seralite® assay (Abingdon Health/Sebia, Evry, France) enables simultaneous determination of kappa and lambda FLCs using a competitive inhibition mechanism. Each test strip incorporates immobilized FLCs that compete with patient FLCs for binding to gold nanoparticle-conjugated monoclonal antibodies. The signal, read by the ADxLR5 device, is inversely proportional to analyte concentration [40]. This design reduces the risk of antigen excess effects. No significant differences are found in median FLC values compared to the Freelite® test, although the kappa/lambda ratio of Seralite® has a wider range [40]. Hence, Seralite® is quantitative, with better performance in high-concentration settings, but interpretation of ratios may differ.

The Sebia enzyme-linked immunosorbent assay (ELISA) (Sebia, Evry, France) uses a sandwich format with rabbit polyclonal antibodies, with the secondary antibody labeled by horseradish peroxidase [41]. It has been fully automated on the AP22 ELITE processor. The assay correlates well with Freelite® and, due to its broader dynamic range, reduces the need for repeat dilutions. However, biases emerge at higher concentrations [42]. This approach therefore provides robust quantitative measurement with improved linearity compared to other immunoassays.

Overall, serum FLC measurement significantly increases sensitivity in screening panels for AL amyloidosis, leading to its inclusion in IMWG recommendations since 2009 [43], [44].

Despite improvements, immunoassays remain prone to technical limitations such as the hook effect. Alternative technologies are being explored to improve precision. Mass spectrometry (MS) has become particularly attractive with advances in accessibility and instrumentation. By targeting the unique amino acid sequence in the complementarity-determining region (CDR), MS can provide both identification and accurate quantification of monoclonal light chains.

The clonotypic peptide approach requires prior sequencing of the clonal light chain to monitor unique peptides by LC–MS/MS [45], [46]. Although technically demanding, once established it offers highly sensitive and accurate quantification, particularly for measurable residual disease. By contrast, the miRAMM (monoclonal immunoglobulin rapid accurate mass measurement) technique does not require prior sequence knowledge [47]. Immunoglobulins were first enriched from samples, then reduced to release free light chains, which were subsequently subjected to LC-ESI-TOF MS analysis. The monoclonal light chain appeared as a single peak when the mass spectra of multiply charged light chains were deconvoluted to determine their molecular mass. The distinct m/z distribution of their constant sections allowed for the differentiation of kappa and lambda light chains. Without requiring chromatography, the same team that created this initial technology modified the miRAMM methodology to a MALDI-TOF MS (matrix-assisted laser desorption ionization time of flight MS) [48]. These techniques are more sensitive than protein electrophoresis or immunofixation at identifying the monoclonal light chain, albeit their sensitivity varies with the quantity of polyclonal background; in addition to identification, the size of this peak can be tracked quantitatively over time, providing a reproducible measure of disease burden with greater sensitivity than electrophoresis or immunofixation  [49].

Research on these methods is ongoing, and the IMWG Mass Spectrometry Committee, based on available evidence in 2021, issued recommendations regarding their use in plasma cell disorders [50]. They stated that mass spectrometry can be used as an alternative to immunofixation in clinical patient management but did not endorse its use as a replacement for the current immunochemical measurement of FLCs in amyloidosis and in other plasma cell disorders.

Lines 288-330: The primary and most ancient method for identifying amyloid fibril deposits on formalin-fixed and paraffin-embedded (FFPE) tissue sections is the Congo red staining, which dates back to 1922. Using a polarized light microscope, amyloid is shown as green birefringent deposits. Using an alkaline staining solution that is saturated with sodium chloride has further enhanced this technique. Tissue amyloid can also be found with metachromatic stains like crystal violet and fluorescent stains like thioflavin T or S.

Amyloid proteins on tissues are mostly identified and characterized by immunohistochemistry, either in electron microscopy (on glutaraldehyde-fixed tissue samples) using post-embedding immunogold techniques or in light microscopy (on FFPE tissues) using immunoperoxidase techniques [56]. Despite the limitations (i.e., sensitivity) and potential drawbacks (see specificity) of antibody-based techniques, MS has been promoted as a viable means of identifying the type of amyloid fibrils. Four basic steps serve as the foundation for the two primary MS techniques: first, the sample's proteins are all broken down, commonly using trypsin. The next step is to separate the pieces of 5–25 amino acids (aa); liquid chromatography (LC) is the most often used method. The peptides are then ionized by subjecting the solution to high voltages. Tandem MS/MS analysis includes a mass/charge (m/z) measurement, peptide fragmentation upon contact with an inert gas, a technique known as collision induced fragmentation (CID), and a final MS measurement of the peptide's unique CID, which enables the identification of the precise amino acid sequence. Using bioinformatic methods, the results are compared with reference databases in the last phase to ascertain the likelihood that each peptide comes from a particular protein [57].

A new technique is to use a fluorescence-equipped microscope to segment 10 µm tissue slices from FFPE specimens and isolate amyloid deposits using laser microdissection (LMD). Amyloid deposits may be effectively separated from the background with LMD, providing material for LC-MS/MS analysis and bioinformatic analysis (sensitivity and specificity of 98–100%) [57].

An alternative approach has been proposed [52], involving shotgun LC-MS/MS analysis to generate proteome maps of unfractionated tissue (typically fresh fat, although FFPE samples can also be used). Amyloid-positive samples are compared to negative control tissues, and amyloid identification is based on the alpha-value [52], a parameter quantifying the relative abundance of known amyloid proteins in patients versus controls.

The lack of knowledge regarding the fibrils' spatial distribution is the main limit of both these latter options. The developement of imaging-assisted MS was a recent solution to this problem. To distinguish ATTR from AL-lambda amyloidosis, matrix assisted laser desorption/ionization mass spectrometry imaging combined with ion mobility separation (MALDI-IMS MSI) has a 91% sensitivity and a 94% specificity [58].

Using shotgun proteomic techniques to analyze the entire tissue, amyloidogenic fibrils can be identified only by the presence of specific proteins that often deposit with them: Serum amyloid P, apolipoprotein IV, and apolipoprotein E [19].

Lines 363-373: Serum FLCs are crucial tools for identifying AL amyloidosis, determining the patient risk, and directing the treatment. There are five assays available. Depending on whether the FLCs are found in monomeric or dimeric forms, they may produce different outcomes. They are based on antigen-antibody recognition. The main drawbacks are that the two most commonly used assays yield differing results, and that even the Freelite® assay might produce inconsistent outcomes depending on the platform and methodology employed. A tissue biopsy is mandatory for the diagnosis of AL amyloidosis, and AL amyloid must be identified using immunohistochemistry and potentially proteomic techniques. Upon diagnosis of AL amyloidosis, precise risk stratification is essential for guiding treatment decisions and depends critically on circulating FLC levels. Lastly, circulating FLCs variations over time are significant markers of treatment response.

  • The topic of the review is relevant, but the number of literature references (60) is relatively small for a literature review.

We thank the Reviewer for the note. We have added the following references:

Baden, J Biol Chem. 2009

Bochtler et al., Haematologica, 2008

Caponi L. Critical Reviews in Clinical Laboratory Sciences. 2020

Daves et al., Biochemia Medica. 2021

Fleming et al., Clin Chem Lab Med. 2019

Gertz et al., 2020, JAMA

Gottwald J. Crit Rev Biochem Mol Biol. 2021

Kaplan et al., Clinica chimica acta, 2020

Merlini & Palladini, Ann Oncol, 2008)

Milani P et al., Blood (2017)

Morgan et al. Amyloid, 2025

Murphy, C. L., et al. American Journal of Clinical Pathology 2001

Rezk, T., et al. Journal of Pathology: Clinical Research, 2019.

Sidana, Leukemia. 2018

Wechalekar et al., 2017, Br J Haematol

  • Figure 3 is presented at insufficient resolution, making the text difficult to read.

We thank the Reviewer for the consideration. We have provided to obtain a simplified version with a better resolution.

Round 2

Reviewer 2 Report

Comments and Suggestions for Authors

The corrections, changes and additions are satisfactory 

Author Response

Thank you very much for your review and constructive comments.

Reviewer 4 Report

Comments and Suggestions for Authors

The authors have taken many of the comments into account and revised the manuscript, and it now resembles their previous publication to a lesser extent (Camerini L, Aimo A, Pucci A, Castiglione V, Musetti V, Masotti S, Caponi L, Vergaro G, Passino C, Clerico A, Franzini M, Emdin M. Serum and tissue light-chains as disease biomarkers and targets for treatment in AL amyloidosis. Vessel Plus. 2022;6:59. http://dx.doi.org/10.20517/2574-1209.2022.06). Unfortunately, some fragments of the manuscript still repeat text from the work of Camerini L. et al., 2022. For example, the following text remains largely identical between the two works: “The advent of automated serum FLC assays has markedly improved the quantification of serum FLC levels, the kappa/lambda ratio and the ratio between involved and uninvolved FLC (i/uFLC), thereby revolutionizing the diagnosis and monitoring ofplasma cell disorders [32]. In subjects with normal kidney function, the kappa/lambda ratio ranges between 0.26 and 1.65 when using the Freelite® assay. In patients with chronic kidney disease, the reticuloendothelial system becomes more important for FLCremoval, and the ratio can increase up to 3 with the same assay (range 0.37-3.10) (Figure2) [33]. In patients with cardiac disease, a normal kappa/lamba FLC ratio has a 100% neg-ative predictive value for AL-CA [34].” (Pages 4-5). Also, the text: “Five commercial diagnostic assays are currently available for measuring serum FLC:Freelite®, N Latex FLC, Diazyme, Seralite®, and Sebia FLC assays. Each method is basedon antibodies able to identify specific epitopes of the FLC. These are described as “hidden” because they seem to be visible only when the LCs are actually free, while they cannot berecognised when they are paired with heavy chains.” (Page 6).
Please include a reference to the article (Camerini L, Aimo A, Pucci A, Castiglione V, Musetti V, Masotti S, Caponi L, Vergaro G, Passino C, Clerico A, Franzini M, Emdin M. Serum and tissue light-chains as disease biomarkers and targets for treatment in AL amyloidosis. Vessel Plus. 2022;6:59. http://dx.doi.org/10.20517/2574-1209.2022.06), since material from it is used in the manuscript. At present, this article is still not cited in the text or included in the references. This concern remains unaddressed.

In addition, there are other reasons not to recommend the article for publication, which are listed below:

The authors have incorporated some new literature into the manuscript. However, it must be noted that the topic of the review is highly relevant and new important papers appear every year. At the same time, this manuscript contains only one reference to a work published later than 2022 (Morgan et al. Amyloid, 2025), which again does not make it substantially better than the authors’ earlier work (Camerini L. et al., 2022). To enhance the relevance of the review, please add consideration and discussion of multiple works published after 2022, for example:

Wang, Y., Liu, F., Liu, Y. et al. Immunoglobulin heavy/light chain assay in the diagnosis, monitoring and follow-up of renal AL amyloidosis patients at different disease stages. Ann Hematol 104, 2287–2295 (2025). https://doi.org/10.1007/s00277-025-06345-7

Gonzalez-Hernandez, D.R., Hanouneh, M. & Cervantes, C.E. An atypical cause of amyloidosis: a case of combined heavy and light chain amyloidosis. BMC Nephrol 26, 332 (2025). https://doi.org/10.1186/s12882-025-04226-9

Baur, J., Berghaus, N., Schreiner, S., Hegenbart, U., Schönland, S. O., Wiese, S., … Haupt, C. (2022). Identification of AL proteins from 10 λ-AL amyloidosis patients by mass spectrometry extracted from abdominal fat and heart tissue. Amyloid, 30(1), 27–37. https://doi.org/10.1080/13506129.2022.2095618

Joshua Bomsztyk, Sriram Ravichandran, Hannah V. Giles, Nicola Wright, Oscar Berlanga, Jahanzaib Khwaja, Shameem Mahmood, Brendan Wisniowski, Oliver Cohen, Darren Foard, Janet Gilbertson, Muhammad U. Rauf, Neasa Starr, Ana Martinez-Naharro, Lucia Venneri, Carol Whelan, Marianna Fontana, Philip N. Hawkins, Julian D. Gillmore, Helen Lachmann, Stephen Harding, Guy Pratt, Ashutosh D. Wechalekar; Complete responses in AL amyloidosis are unequal: the impact of free light chain mass spectrometry in AL amyloidosis. Blood 2024; 143 (13): 1259–1268. doi: https://doi.org/10.1182/blood.2023022399

The figures in the manuscript generally help to attract readers’ attention. However, I have the following questions and comments regarding them:

Figure 1. Mechanisms of damage for free light chains. ─ How does this figure illustrate the mechanisms of damage? How are the damages themselves shown? In its current form, the figure is not very informative.

Figure 2. The authors have, for some reason, repeated the information on the kappa/lambda ratio ranges 0.26–1.6 and 0.37–3.10 multiple times – in the figure, in the figure legend, and in the text preceding the figure. What new information does this figure provide compared to the text? Please make Figure 2 more informative. For example, add visual information comparing the Freelite® assay (The Binding Site, UK) and the N Latex FLC assay (Siemens Healthineers).

Figure 3. Please spell out the abbreviation PSM in the figure legend.

Legend to Figure 3. “D) The most abundant amyloidogenic protein (highlighted in orange) is identified, while co-precipitating proteins (in yellow) represent part of a characteristic amyloid protein signature.” ─ In the current version of the manuscript, there are no color markings.

There are also problems with the structure of the text. For example:

Page 1. The text begins without a heading: “ Amyloidosis is a disease…” ─ Is this the Introduction section? If so, please label it accordingly.

Page 6. Table 1. The abbreviations “ELISA, enzyme-linked immunosorbent assay; FLC, free light-chain;…” appear to belong beneath the table. Please check.
